# Sparsified State-Space Models are Efficient Highway Networks

**Woomin Song**                                                                 *woomin.song@kaist.ac.kr*
*Korea Advanced Institute of Science & Technology (KAIST)*

**Jihoon Tack**                                                                    *jihoontack@kaist.ac.kr*
*Korea Advanced Institute of Science & Technology (KAIST)*

**Sangwoo Mo**                                                                           *swmo@umich.edu*
*University of Michigan, Ann Arbor*

**Seunghyuk Oh**                                                               *seunghyukoh@kaist.ac.kr*
*Korea Advanced Institute of Science & Technology (KAIST)*

**Jinwoo Shin**                                                                        *jinwoos@kaist.ac.kr*
*Korea Advanced Institute of Science & Technology (KAIST)*

**Reviewed on OpenReview:** *https://openreview.net/forum?id=G1pOYwrX8X*

## Abstract

State-space models (SSMs) offer a promising architecture for sequence modeling, providing an alternative to Transformers by replacing expensive self-attention with linear recurrences. In this paper, we propose a simple yet effective trick to enhance SSMs within given computational budgets by sparsifying them. Our intuition is that tokens in SSMs are highly redundant due to gradual recurrent updates, and dense recurrence operations block the delivery of past information. In particular, we observe that upper layers of SSMs tend to be more redundant as they encode global information, while lower layers encode local information. Motivated by this, we introduce Simba, a hierarchical sparsification method for SSMs based on token pruning. Simba sparsifies upper layers more than lower layers, encouraging the upper layers to behave like highways. To achieve this, we propose a novel token pruning criterion for SSMs, measuring the global impact of tokens on the final output by accumulating local recurrences. We demonstrate that Simba outperforms the baseline model, Mamba, with the same FLOPS in various natural language tasks. Moreover, we illustrate the effect of highways, showing that Simba not only enhances efficiency but also improves the information flow across long sequences. Code is available at `https://github.com/woominsong/Simba`.

## 1 Introduction

State-space models (SSMs) (Gu et al., 2022b; Gu & Dao, 2023) offer a promising architecture for sequence modeling, efficiently handling sequences using linear recurrence structures. Thanks to this efficiency, SSMs have shown potential as an alternative to Transformers (Vaswani et al., 2017), which use the self-attention mechanism, incurring high computational costs for long sequences. In particular, Mamba (Gu & Dao, 2023) has recently demonstrated that SSMs can scale up to billions of parameters and show comparable performance with Transformers in various domains (Zhu et al., 2024; Li et al., 2024a;b).

After their success, numerous works have aimed to enhance SSMs and Mamba further. One popular approach involves hybrid models combining Transformers and SSMs (Lieber et al., 2024; Poli et al., 2024). This approach assists the models in retaining past information through the global memory of Transformers (Vardasbi et al.,

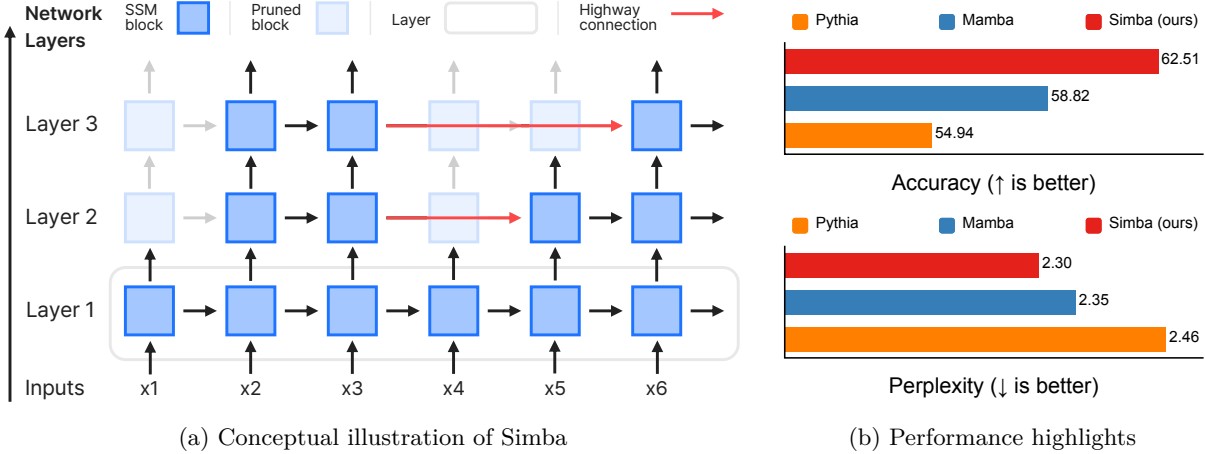

(a) Conceptual illustration of Simba      (b) Performance highlights

Figure 1: **Simba: hierarchical sparsification of SSMs via token pruning.** (a) We found that tokens in state-space models (SSMs) are highly redundant, especially in the upper layers. Motivated by this observation, we propose hierarchically sparsifying pre-trained SSMs by progressively pruning tokens across layers. This results in models with a trapezoidal shape, featuring sparse upper layers that act like highways, enhancing efficiency and information flow of the original SSM. (b) We highlight results comparing Simba-2.8b with Mamba and Pythia, all with the same number of FLOPS. We report the mean accuracy over 6 NLP benchmarks and perplexity on the PG-19 dataset with 2k context, following the setups in Section 4. Simba outperforms both models in accuracy and perplexity.

2023; Jelassi et al., 2024). However, this compromises the efficiency of SSMs by reintroducing expensive self-attention. Instead of sacrificing efficiency, we explore an alternative direction to improve SSMs by comparing models on a fixed computational budget. Specifically, we investigate sparsification of large pre-trained SSMs, known to yield better models than training small ones from scratch (Frankle & Carbin, 20189).

To this end, we first analyze the behavior of tokens in pre-trained SSMs. We observe that tokens in SSMs are highly redundant, as they are gradually updated over sequences. This redundancy is particularly noticeable in the upper layers. Furthermore, dense recurrence operations over the redundant tokens block the delivery of past information, potentially harming the contextual understanding of SSMs.

Inspired by this, we propose Simba, a simple yet effective method to sparsify SSMs through token pruning (i.e., it is training-free). Our core idea is to sparsify SSMs into a hierarchical form, enforcing more sparsity in upper layers than in lower layers. As a result, the upper layers behave as highways to transmit past information, enabling efficient inference and facilitating the information flow across long sequences. Figure 1a illustrates the visual concept of our approach, obtaining a trapezoidal-shaped sparsified network.

To implement this, we propose a novel token pruning criterion for SSMs. Specifically, our score measures the global influence of each token on the final output by reformulating SSM equations to accumulate the effect from local recurrences. This approach can also be viewed as an SSM extension of attention-based token pruning criteria used for Transformers (Goyal et al., 2020). We found that our criterion outperforms intuitive baselines, such as uniform pruning of tokens with even intervals.

Our experiments show that Simba, obtained by sparsifying Mamba without any fine-tuning, significantly outperforms Mamba using the same number of FLOPS in various tasks. For instance, Simba consistently achieves better FLOPS-accuracy curves on 6 NLP benchmarks, including Lambada (Paperno et al., 2016), HellaSwag (Zellers et al., 2019), PIQA (Bisk et al., 2020), ARC-Challenge (Clark et al., 2018), ARC-Easy (Clark et al., 2018), and WinoGrande (Sakaguchi et al., 2021). Here, Simba obtained from Mamba-2.8b performs on par with the original Mamba-2.8b, despite using similar FLOPS to Mamba-1.4b, as highlighted in Figure 1b. For example, it achieves an average accuracy of 62.5% for 6 downstream NLP tasks, improving 58.8% of Mamba-1.4b.

We also demonstrate the language modeling ability of Simba by measuring perplexity on the PG-19 dataset (Rae et al., 2019) across different context lengths. Like the NLP benchmarks, Simba achieves better perplexity than Mamba using the same number of FLOPS. More importantly, Simba performs robustly over long sequences exceeding the pre-trained context length, such as twice longer than the trained length, unlike Mamba, which significantly deteriorates with length extrapolation. This supports the idea that the highway structures in Simba facilitate long sequence modeling.

We further investigate the effect of highways in Simba. Somewhat unexpectedly, we found that Simba performed better than its original unpruned Mamba in some of our experiments, potentially benefiting from the highways created at the upper layers. To further investigate the positive effect of highways, we examine the information flow across layers by assessing the influence of the sequence tokens on the final output. We observe that Mamba relies on tokens near the end across all layers, while Simba also focuses on earlier tokens at the upper layers, showcasing the role of highways.

## 2 Related work

**State-space models** (SSMs) are a powerful architecture for sequence modeling, integrating concepts from classic control theory (Kalman, 1960) with recurrent neural networks (Elman, 1990). The key idea of SSMs is to employ linear recurrence (Katharopoulos et al., 2020; Gu et al., 2022b;a; Mehta et al., 2023; Smith et al., 2023; Fu et al., 2023; Orvieto et al., 2023; Poli et al., 2023; Peng et al., 2023; Sun et al., 2023; De et al., 2024), enabling efficient parallel inference and effective training, unlike Transformers using self-attention (Vaswani et al., 2017), which requires quadratic computation over the sequence length. As a result, SSMs have shown success in handling long sequences (Tay et al., 2021). Recently, Mamba (Gu & Dao, 2023) further scaled up SSMs through a selection mechanism and hardware-aware algorithm, showing the potential of SSMs in challenging tasks such as language, audio, and video (Zhu et al., 2024; Li et al., 2024a;b). We aim to further improve Mamba through network sparsification, efficiently utilizing a fixed computational budget.

**Sparsifying networks** have been widely studied, primarily for training efficient models (Han et al., 2016). Most prior work focused on weight pruning, which removes unnecessary edges in weight matrices (Zhu & Gupta, 2017; Gale et al., 2019; Park et al., 2020; Lee et al., 2021). However, while weight pruning reduces model size, it does not enhance inference speed due to the batch computation nature of GPUs. To address this, structured pruning removes entire blocks at once, such as channels in CNNs (Li et al., 2017) or attention heads in Transformers (Michel et al., 2019). On the other hand, some works focused on the performance benefits of sparsified networks, suggesting that sparsifying large networks yields better models than training small models from scratch, known as lottery tickets (Frankle & Carbin, 20189; Li et al., 2020). Our work relates to structured pruning, as removing tokens speeds up computation and reduces memory usage, and to lottery tickets, as it also improves performance.

**Token pruning** (or merging) is widely applied in Transformers to reduce heavy computation over long sequences (Goyal et al., 2020; Kim et al., 2022; Liu et al., 2022; Bolya et al., 2023; Ke et al., 2024; Shang et al., 2024). In particular, HOMER (Song et al., 2024) has shown that hierarchical token pruning not only reduces computation but also enhances long context understanding by condensing global information into sparse tokens in the upper layers. Our work shares a similar spirit with HOMER but has notable differences. First, we explore token pruning for SSMs, unlike prior works focused on Transformers. To this end, we propose a novel token pruning criterion based on the global importance of tokens to the final output, derived from reformulating SSM equations to accumulate local recurrences. Second, by targeting SSMs, our hierarchical pruning scheme offers a novel interpretation of highway networks, connecting SSMs with classical recurrent networks with long-term memory. As a result, our token pruning approach enhances both the inference efficiency and information flow of SSMs.

**Highway networks** have been proposed to bypass information loss from dense computation through local residuals (He et al., 2016) or long skip connections (Srivastava et al., 2015; Zilly et al., 2017; Huang et al., 2017; Ronneberger et al., 2015). Highways, also termed long-term memory, were used as a standard approach for sequence modeling before global computation methods like self-attention gained popularity (Hochreiter & Schmidhuber, 1997; Chung et al., 2014; Feichtenhofer et al., 2019). However, integrating highways with SSMs proved challenging due to the linear recurrence structure of SSMs, not easily combined with skip connections.

Instead of explicitly using such modules in SSM architectures, we introduce a simple and effective way to integrate highways by pruning dense token connections from the upper layers.

**Tree RNNs** have been explored for processing hierarchical data structures, such as word, sentence, and paragraph hierarchies in language (Hihi & Bengio, 1995; Wang et al., 2019). Despite aligning with human intuition, most methods were unsuccessful due to complex architectures, while simple linear sequence modeling demonstrates its power (Achiam et al., 2023). Our token pruning naturally incorporates this hierarchical structure into SSMs, while favoring the success and scalability of linear sequence modeling.

## 3 Simba: Hierarchical sparsification for state-space models

In Section 3.1, we review the mathematical formula of state-space models (SSMs) and discuss observations on the token redundancy of Mamba. In Section 3.2, we describe our proposed hierarchical sparsification approach for SSMs and explain the token pruning criteria.

### 3.1 Motivation: Hierarchy in SSM token redundancy

**Structured state-space model** (S4) (Gu et al., 2022b) is a family of recently proposed SSMs. In its continuous form, the SSM updates the state $h(t)$ using the input $x(t)$ and produces the output $y(t)$ according to Eq. (1) where $A$, $B$, and $C$ refer to the parameters of SSMs. It discretizes the update rules for discrete sequences, as shown in Eq. (2).

$$h'(t) = Ah(t) + Bx(t), \quad y(t) = Ch(t) \tag{1}$$

$$h_t = \bar{A}h_{t-1} + \bar{B}x_t, \quad y_t = Ch_t \tag{2}$$

Mamba (Gu & Dao, 2023) further improves this formulation using an input selectivity mechanism, creating the matrices $\bar{A}$, $\bar{B}$, and $C$ dependent on the input $x_t$. Thus, the state update equation can be rewritten as follows, where $\bar{A}_t$, $\bar{B}_t$, and $C_t$ denote the input-dependent matrices created using $x_t$.

$$h_t = \bar{A}_t h_{t-1} + \bar{B}_t x_t, \quad y_t = C_t h_t \tag{3}$$

**Hierarchy in token redundancy.** As the state update depends solely on the input and the immediate previous states, the model must compress all previous information into the state of each token. Thus, the states of SSM tokens are likely highly redundant, as states at similar positions would compress a similar set of information. To verify this, we visualize the token redundancy of Mamba in Figure 2 by measuring the cosine similarity between adjacent tokens across the network layers.

Our analysis indicates that tokens in the upper layers tend to exhibit more redundancy than those in the lower layers. One possible explanation is that SSMs process information in a hierarchical manner: lower layers focus on local information, while upper layers focus on global information. This aligns with findings from attention visualizations of Mamba models Ali et al. (2024), where lower layers emphasize diagonal elements in the attention map, while upper layers highlight lower-triangular elements. Since upper layer states encapsulate global information, states at similar positions contain similar information, resulting in higher redundancy.

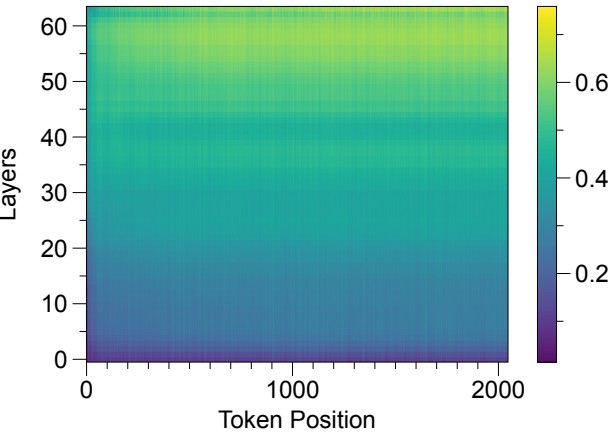

Figure 2: **Token redundancy of SSMs has a hierarchical structure.** We measure the cosine similarity between adjacent tokens of the Mamba-2.8b model across layers, averaged over documents from the PG-19 test dataset. The tokens are highly redundant, especially in the upper layers.

## 3.2 Hierarchical sparsification for SSMs

We introduce Simba, a simple yet effective sparsification approach that can be directly applied to any pre-trained state-space models (SSMs) in a plug-and-play manner. This introduces an efficient highway for effective sequential modeling. Based on prior motivation, Simba sparsifies the full SSM in a hierarchical manner: it sparsifies redundant tokens in upper layers while preserving the local information captured in lower layers, guided by our novel token importance criteria.

**Hierarchical sparsification through token pruning.** To achieve sparsity in SSMs in a hierarchical manner, we propose token pruning at each layer with a specified rate. Implementing this pruning technique at a particular layer automatically decreases the computational burden on subsequent layers, as pruned tokens are no longer propagated to the next layer. Consequently, by sequentially applying token pruning from lower to upper layers, the pruning ratio consistently increases during propagation, thereby introducing hierarchical sparsification to SSMs.

**Token pruning criterion.** We propose a novel token importance score for SSMs to prune the least important token from each layer. To achieve this, we leverage the influence that each token has on the final token's output. This calculation is straightforward due to the linear recurrence property of SSMs. Formally, for a given token sequence $(x_1, \cdots, x_T)$ of length $T$ and the final token output $y_T$ of the layer, we estimate the influence of token $x_t$ at position $t$ by considering the updated output $y_T^{(t)}$ (obtained by removing $x_t$ from $y_T$), as follows:

$$
\begin{aligned}
\Delta y_T(t) &:= y_T - y_T^{(t)} \\
&= \sum_{r=1}^{T} C_T \left( \prod_{k=r+1}^{T} \bar{A}_k \right) \bar{B}_r x_r - \sum_{r=1, r \neq t}^{T} C_T \left( \prod_{k=r+1}^{T} \bar{A}_k \right) \bar{B}_r x_r \\
&= C_T \left( \prod_{k=t+1}^{T} \bar{A}_k \right) \bar{B}_t x_t.
\end{aligned}
\tag{4}
$$

Here, we found that the influence measure $\Delta y_T(t)$ aggregated with the max pooling (or the $\ell_2$ norm) serves as an effective pruning criterion, denoted as $s(t) := \max(\Delta y_T(t))$; we use max pooling throughout the paper as it was slightly better than $\ell_2$ norm. Also, we empirically observed that excluding the bias term when computing $\bar{A}_k$ in fully connected layers was beneficial, as it potentially helps normalize token importance scores across different channels before applying max pooling, resulting in more consistent pruning decisions. Based on the proposed score $s(t)$, we prune each layer by removing the tokens with the lowest scores to obtain a sparsified SSM. The pruning process is performed per input sequence during inference and is based solely on token importance scores computed for that specific input. As a result, no external pruning data is required to determine the pruning pattern.

**Pruning schedule.** The pruning schedule aims to balance the trade-off between efficiency and performance. Upper layers, with their ability to model global context, are better equipped to identify important tokens. Conversely, pruning tokens at earlier layers offers more significant computational savings. Therefore, we propose a linear pruning schedule, inspired by prior work on Transformers (Bolya et al., 2023), where the number of active tokens is linearly reduced across all layers, resulting in a trapezoidal-shaped network after sparsification.

**Highways in sparsified SSMs.** Long-term dependency is a well-known challenge for recurrent models (Hochreiter & Schmidhuber, 1997). Dense recurrence operations tend to attenuate previous information, restricting the information flow across distant tokens. Our sparsification scheme addresses this issue by reducing the number of recurrence operations. As unimportant tokens are pruned during sparsification, upper layers can selectively focus on processing more important information without being burdened by dense recurrence operations on redundant tokens. This highway effect enables our pruning scheme to not only enhance inference efficiency but also facilitate the information flow across distant tokens (Section 4.3).

## 4 Experiments

In this section, we demonstrate the performance of Simba on diverse tasks. In Section 4.1, we evaluate its performance on 6 NLP benchmarks, consistently showing superior performance compared to dense models with equivalent computational resources. In Section 4.2, we assess the language modeling ability of Simba by measuring perplexity conditioned on various context lengths. In Section 4.3, we further investigate the highway effects of Simba. Finally in Section 4.4, we perform ablation studies on token pruning criteria and different pruning ratios, along with a simple fine-tuning experiment.

**Common setups and baselines.** We primarily apply our sparsification method to pre-trained Mamba models of various scales. For our method, we implement a linear pruning schedule that preserves 10% of the tokens at the final layer unless specified otherwise. The Simba models are generated by sparsifying Mamba models without any fine-tuning, following a plug-and-play approach. We conduct performance comparisons of Simba against Mamba (Gu & Dao, 2023) and Pythia (Biderman et al., 2023) models that utilize a similar amount of computation. In NLP tasks consisting of a prompt and a label, we exclusively apply sparsification to the prompts to ensure an accurate evaluation of the label logits. The final token of the prompt remains unpruned, as its output is utilized for computing the label logits. Moreover, to accommodate the task structure, the token importance score is computed with respect to the final token of the prompt. For additional details, refer to Appendix A.

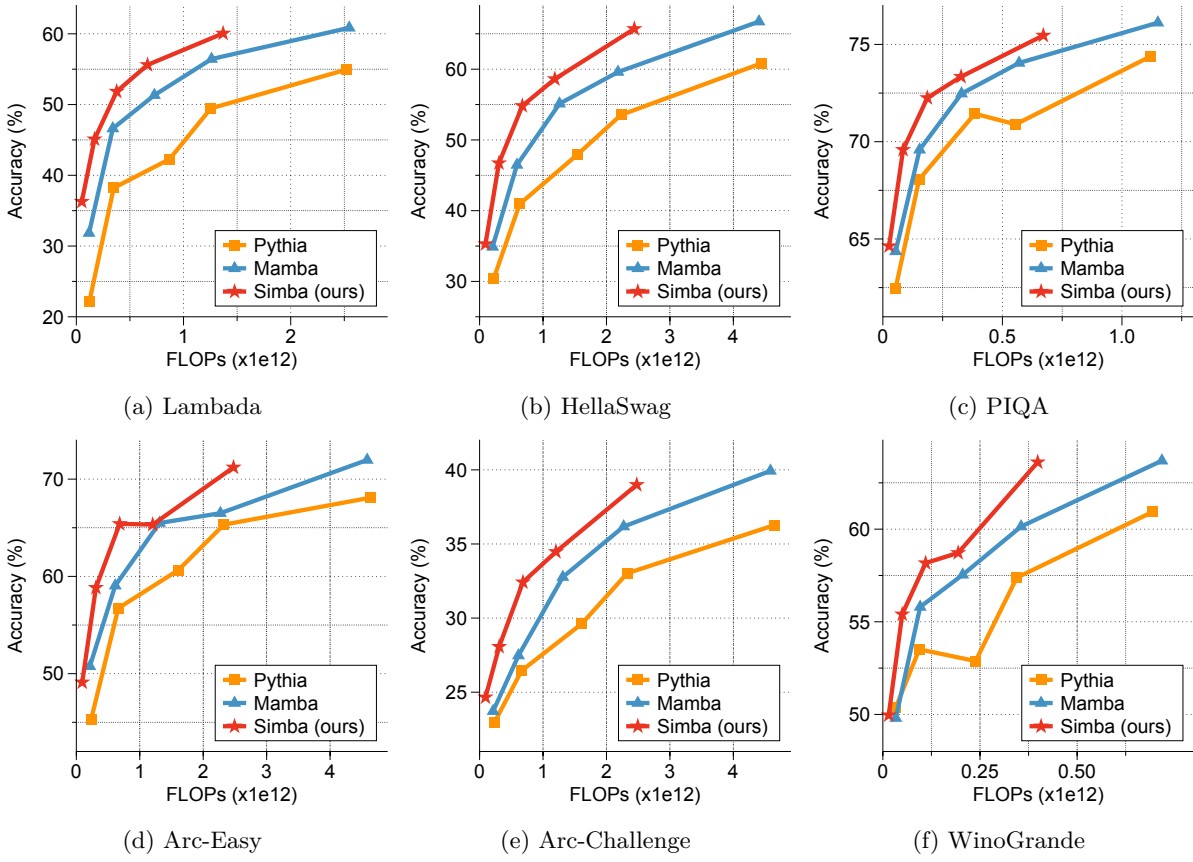

Figure 3: **Performance on NLP Benchmarks.** We visualize the FLOPs-accuracy curve of Mamba, Pythia, and Simba models of various scales on 6 NLP benchmarks. Across all benchmarks, Simba consistently outperforms the baselines using the same number of FLOPs.

## 4.1 NLP benchmarks

In this section, we assess the language understanding capability of Simba by evaluating six downstream NLP tasks. Specifically, we present the performance and computational efficiency of Simba on the Lambada (Paperno et al., 2016), HellaSwag (Zellers et al., 2019), PIQA (Bisk et al., 2020), ARC-Challenge (Clark et al., 2018), ARC-Easy (Clark et al., 2018), and WinoGrande (Sakaguchi et al., 2021) benchmarks. Consistent with Mamba (Gu & Dao, 2023), we report accuracy normalized by sequence length for HellaSwag and ARC-Challenge, and accuracy for the other datasets. All evaluations use the LM evaluation harness from EleutherAI (Gao et al., 2021).

We report evaluation accuracy and computational cost for each benchmark in Figure 3. As evident from the results, Simba provides the best accuracy-efficiency trade-off, consistently outperforming other models with the same number of FLOPs. This demonstrates that Simba can successfully make dense Mamba models sparse, advancing the frontier of the accuracy-efficiency trade-off. We provide the full results for all benchmarks in Table 4 of the Appendix.

## 4.2 Language modeling

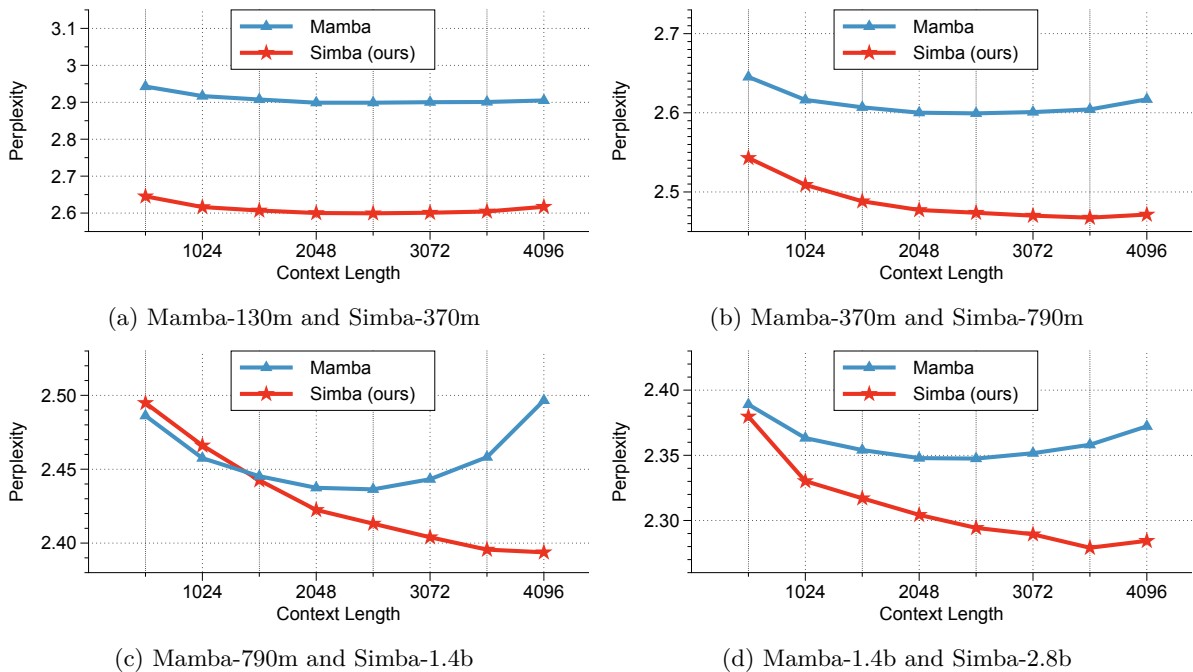

(a) Mamba-130m and Simba-370m

(b) Mamba-370m and Simba-790m

(c) Mamba-790m and Simba-1.4b

(d) Mamba-1.4b and Simba-2.8b

Figure 4: **Language modeling ability.** We measure the FLOPs-perplexity curves on the PG-19 test dataset. Simba models are compared against Mamba models that use similar computation. Simba not only outperforms Mamba with the same computation but also shows decreasing perplexity after its pre-trained context limit of 2k tokens.

In this section, we evaluate the language modeling ability of Simba by measuring perplexity on long documents. Specifically, we measure the perplexity of short document snippets sampled from PG-19 dataset (Rae et al., 2019), conditioned on varying amounts of context. We keep the 100-token snippet fixed for all experiments to ensure that all perplexity measurements are done on the same set of tokens.

We report perplexity values conditioned on varying contexts in Figure 4. The results show that Simba consistently shows improved perplexity with similar computation, outperforming the dense Mamba models. We provide the full results, including Pythia, in Table 4 of the Appendix.

**Long context capability.** An intriguing observation is that Simba, unlike Mamba models, exhibits decreasing perplexity even after surpassing its pre-trained context limit of 2k tokens. Context limits, defining the maximum number of tokens a model can handle, often result in catastrophic performance drop if exceeded (Song et al., 2024). While SSMs are generally more resilient to this issue, the results indicate increasing perplexity for Mamba models with longer contexts, suggesting ineffective utilization of additional context. In contrast, Simba consistently demonstrates decreasing perplexity even with extended contexts, highlighting its adeptness at leveraging extra information.

This benefit is likely attributed to the highways formed in the upper layers due to extensive sparsification. The performance drop with longer inputs typically results from a distribution shift in the training data, as the model never saw the long input length during training. For dense models like Mamba, this shift affects all network layers. Conversely, sparse models like Simba process significantly fewer tokens in the upper layers, thanks to extensive sparsification. Consequently, the upper layers remain unaffected by distribution shifts in input lengths, processing inputs more effectively.

In summary, our experiments demonstrate that Simba not only outperforms Mamba with the same computation but also exhibits superior long-context handling abilities, suggesting the benefits of highways.

### 4.3 Sparsified SSMs as highway networks

In this section, we further investigate the highway effects of Simba. First, we identify some scenarios where Simba models perform better than the original dense models despite using fewer FLOPs, potentially benefiting from the highways. Second, we examine the information flow in the model, showing that highways assist in obtaining information from earlier tokens, unlike dense SSMs, which over-rely on later tokens.

Table 1: **Comparsion between same model scales.** We compare Simba and Mamba on NLP benchmarks: Lambada (Lbd.), HellaSwag (HS), PIQA, Arc-Easy (Arc-E), Arc-Challenge (Arc-C), and WinoGrande (WG). We use a moderate pruning ratio for Simba, leaving 70% of the tokens at the final layer. Bold denotes the best results, showing that Simba often improves Mamba while using fewer FLOPS.

| Model | Scale | Model Dim. | FLOPs (x1e12) | Lbd. acc. (↑) | HS acc. (↑) | PIQA acc. (↑) | Arc-E acc. (↑) | Arc-C acc. (↑) | WG acc. (↑) | Avg. acc. (↑) |
|---|---|---|---|---|---|---|---|---|---|---|
| Mamba | 130m | 768 | 0.48 | 31.84 | 34.88 | 64.36 | **50.76** | 23.72 | 49.80 | 42.56 |
| Simba (ours) | 130m | 768 | **0.39** | **32.43** | **35.05** | **64.42** | 50.38 | **24.32** | **49.88** | **42.75** |
| Mamba | 370m | 1024 | 1.38 | 46.61 | 46.46 | 69.59 | 59.05 | 27.47 | **55.80** | 50.83 |
| Simba (ours) | 370m | 1024 | **1.15** | **47.09** | **46.69** | **69.64** | **59.51** | **27.65** | 55.49 | **51.01** |
| Mamba | 790m | 1536 | 2.94 | 51.33 | **55.12** | **72.47** | 65.49 | **32.76** | **57.54** | 55.78 |
| Simba (ours) | 790m | 1536 | **2.47** | **51.82** | 54.93 | 72.20 | **65.87** | **32.76** | 57.22 | **55.80** |

**Comparison under same model sizes.** In the previous experiments, we mainly compare models with the same number of FLOPS. Here, we provide an additional comparison between the dense and sparse models that have the same scale. Specifically, we evaluate Simba models with a more moderate pruning ratio, using a linear pruning schedule with 70% of tokens remaining at the final layer. We benchmark Mamba and Simba models using the 6 NLP benchmarks, following the setup in Section 4.1. We provide the evaluation results in Table 1.

Somewhat unexpectedly, we found that Simba models sometimes perform even better than the original model, indicating the possible benefits of highways created at the upper layers. The gain is more evident for smaller models, possibly because dense recurrence operations are more harmful to smaller state sizes, so highways provide more benefits.

**Highways facilitate information flow from early tokens.** We investigate how information flows through the dense and sparse SSM layers. Specifically, we measure the influence of tokens at each position on the final token, using our token importance score in Eq. (4) but normalized to equalize the contribution of each input document, i.e., $s(t)/||y_T||_2$. See Appendix A.2 for more details.

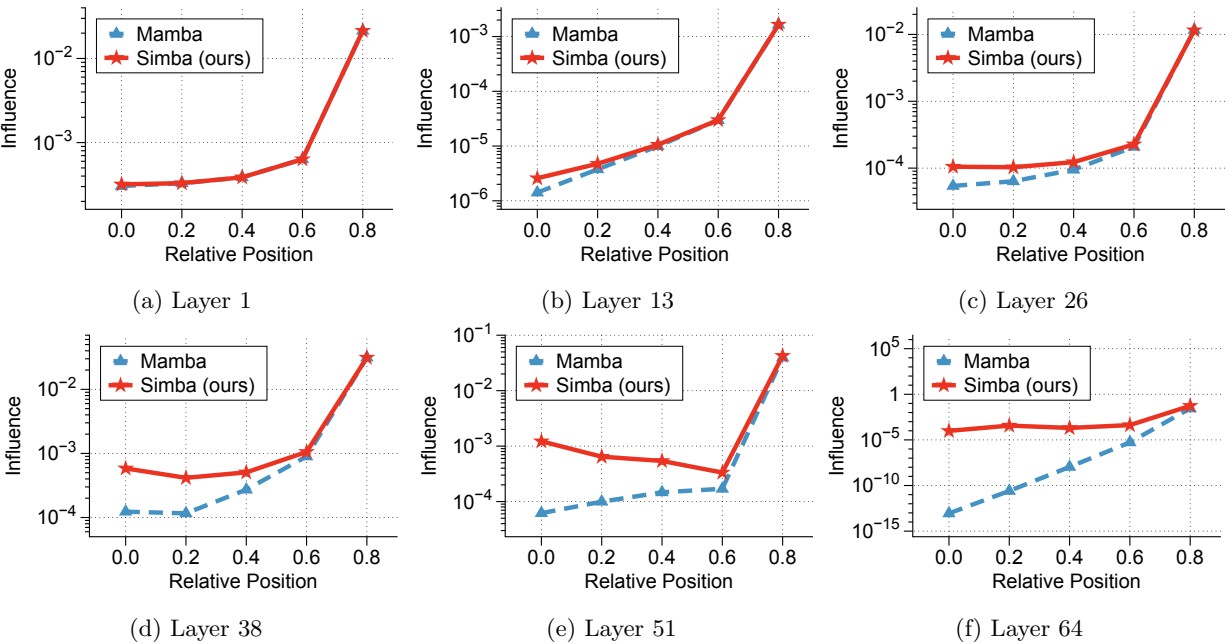

(a) Layer 1      (b) Layer 13      (c) Layer 26

(d) Layer 38      (e) Layer 51      (f) Layer 64

Figure 5: **Information flow across layers.** We visualize the information flow using the normalized token influence score. We compare Mamba-2.8b and Simba-2.8b, averaging scores over the PG-19 test dataset samples. The information flow of Simba flattens at the upper layers, indicating better information flow from early tokens.

We illustrate the results in Figure 5. Mamba demonstrates a consistent information flow pattern across all layers, with tokens near the end exerting more significant influence on the final token. Simba displays a similar trend in the lower layers. However, the slope flattens in the upper layers, indicating that they function as highways, facilitating the flow of information from early tokens.

### 4.4 Ablation study and analysis

This section presents ablation studies on token pruning criteria and different pruning ratios. We also perform a simple fine-tuning experiment to further improve the performance of Simba.

**Pruning criteria.** We compare our proposed pruning criteria in Eq. (4) with two baselines: "Random," which chooses tokens from random positions, and "Uniform," which chooses tokens from evenly distributed intervals. We visualize the efficiency-performance trade-off of Mamba-2.8b and Simba-2.8b models on the Arc-Challenge dataset in Figure 6. Random pruning significantly hurts performance, while Uniform pruning forms a strong baseline, highlighting the necessity of proper token selection. Simba further improves upon Uniform pruning by considering token influence.

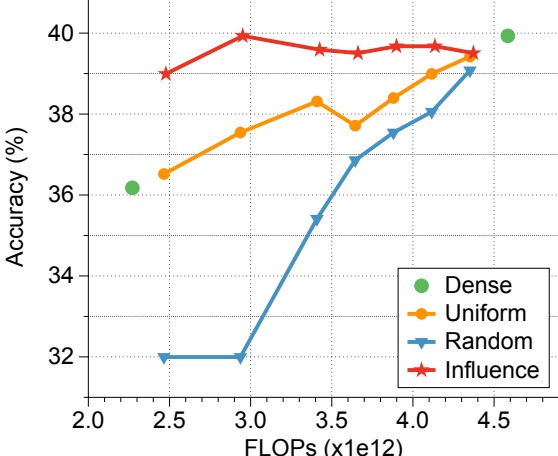

Figure 6: **Ablation study on pruning criteria.** Performance of sparsified Mamba-2.8b models on the Arc-Challenge dataset (Clark et al., 2018) evaluated using different pruning criteria across various pruning ratios. Our proposed token influence score in Eq. (4) performs the best, even remaining robust under severe sparsification. We also report the performance of the dense Mamba-1.4b and Mamba-2.8b models for comparison.

Table 2: **Fine-tuning results.** We compare the language modeling perplexity on the test split of the PG-19 dataset, measured with different context lengths. The fine-tuned Simba model consistently outperforms its training-free counterpart.

| Model | Scale | FLOPs (x1e12) | Within Context | | | | | Extrapolation | | |
|---|---|---|---|---|---|---|---|---|---|---|
| | | | 0.5k | 1k | 1.5k | 2k | 2.5k | 3k | 3.5k | 4k |
| Mamba | 130m | 0.48 | 2.943 | 2.917 | 2.907 | 2.899 | 2.899 | 2.900 | 2.901 | 2.905 |
| Simba (training-free) | 370m | 0.68 | 2.727 | 2.686 | 2.662 | 2.650 | 2.643 | 2.633 | 2.630 | 2.625 |
| Simba (fine-tuned) | 370m | 0.68 | **2.723** | **2.678** | **2.658** | **2.645** | **2.637** | **2.630** | **2.626** | **2.621** |
| Mamba | 370m | 1.38 | 2.645 | 2.616 | 2.607 | 2.600 | 2.599 | 2.601 | 2.604 | 2.617 |

**Pruning ratio.** We compare the performance of Simba using different sparsity levels, with linear pruning schedules leaving 90%, 80%, 70%, 60%, 50%, 30%, and 10% of tokens at the final layer. Figure 6 presents the performance curves for different pruning criteria. Simba is robust to extreme sparsity, retaining performance even when pruning 90% of tokens at the final layer.

**Fine-tuning.** Although our method can be applied to pre-trained SSMs without training, we investigate if the performance can be improved with further fine-tuning. To this end, we perform a simple fine-tuning experiment, further training the Mamba-370m model with MiniPile dataset (Kaddour, 2023), which is a subset of the pre-training dataset (the Pile (Gao et al., 2020)) used for training Mamba. We provide the detailed training configurations in Appendix A.3.

Following the setup in Section 4.2, we evaluate the language modeling perplexity of 100 tokens using the PG-19 dataset, conditioned on varying amounts of context. We report the results in Table 2. For all context lengths, the fine-tuned Simba model consistently outperforms the training-free Simba model, suggesting that fine-tuning can further improve the performance of sparsified SSMs.

## 5  Conclusion

We propose Simba, which sparsifies pre-trained SSMs into a hierarchical form through token pruning. Simba outperforms Mamba with the same number of FLOPS in both accuracy on downstream NLP benchmarks and language modeling perplexity. Additionally, our pruning scheme creates highways in the upper layers, enhancing length extrapolation for long sequences and facilitating the information flow across distant tokens. We hope Simba inspires a broad community, including state-space models, sparse and efficient networks, and classic recurrent networks with highways.

**Limitations.** Our paper mainly focused on applying Simba to the pre-trained Mamba without adjustment. However, token pruning incurs distribution shifts from the original models, and further fine-tuning could reduce this misalignment. We demonstrate that simple fine-tuning can improve the performance of small models within a fixed computational budget in Section 4.4. However, a more sophisticated fine-tuning scheme tailored for sparse SSMs could be investigated. Also, while sparsifying a large model provides high performance at reduced inference-time computation, preparing a large model would require a higher pre-training cost compared to using a smaller, dense model.

**Broader Impact Statement** Our paper studies sequence models, with broad applications such as language, audio, and video generation. As our method enhances the efficiency and efficacy of these models, it holds the potential to impact a broader audience in generative AI. Hence, users of our method and sequence models should carefully read and follow the guidance from the community (Bai et al., 2022).

## 6  Acknowledgements

This work was conducted by Center for Applied Research in Artificial Intelligence(CARAI) grant funded by Defense Acquisition Program Administration(DAPA) and Agency for Defense Development(ADD) (UD230017TD), and partially supported by Institute for Information & communications Technology Pro-

motion(IITP) grant funded by the Korea government(MSIT) (No.RS-2019-II190075 Artificial Intelligence Graduate School Program (KAIST); RS-2022-II220959, Few-shot Learning of Causal Inference in Vision and Language for Decision Making).

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

# A Detailed setups

## A.1 NLP benchmarks

Here, we provide the details for downstream experiments in Section 4.1. The six benchmarks were chosen according to the experiment setup of (Gu & Dao, 2023). The metrics are also selected accordingly, where we measure answer perplexity and accuracy for Lambada, accuracy for PIQA, ARC-Easy, and Winogrande, and normalized accuracy for HellaSwag and Arc-Challenge. All measurements use the LM evaluation harness from EleutherAI (Gao et al., 2021).

Following the widely adopted practice (Beeching et al., 2023), we evaluate the model's downstream performance using few-shot prompts. Following the setup in the open LLM leaderboard (Beeching et al., 2023), we evaluate the model with 10-shot prompts for HellaSwag, 25-shot prompts for ARC-Easy and Arc-Challenge, and 5-shot prompts for WinoGrande. For benchmarks not considered in the leaderboard, we evaluate the models with 5-shot prompts.

For the visualization in Figure 3, we measure FLOPs separately for each benchmark and each model. First, we measure each benchmark's mean prompt and answer length, as shown in Table 3. Then, we measure FLOPs by forwarding an input that matches the mean lengths.

Table 3: **Downstream task details.** We provide the details of our downstream evaluations in Section 4.1, including the number of few-shot prompts, average prompt lengths, average answer lengths, and metrics used for evaluation.

| Dataset | Few-shot Prompts | Prompt Length (avg.) | Answer Length (avg.) | Metrics |
|---|---|---|---|---|
| Lambada (Paperno et al., 2016) | 5 | 507.44 | 1.47 | ppl./ acc. |
| HellaSwag (Zellers et al., 2019) | 10 | 877.25 | 29.86 | acc_norm |
| PIQA (Bisk et al., 2020) | 5 | 229.53 | 22.82 | acc. |
| ARC-Easy (Clark et al., 2018) | 25 | 913.81 | 5.00 | acc. |
| ARC-Challenge (Clark et al., 2018) | 25 | 913.81 | 5.00 | acc_norm |
| WinoGrande (Sakaguchi et al., 2021) | 5 | 143.17 | 5.67 | acc. |

## A.2 Information flow visualization

Here, we provide the details for our information flow visualization experiments in Figure 5. We measure the normalized token influence score $s(t)/\|y_T\|_2$ across all documents from the PG-19 test set, truncated at 1000 tokens. For all samples, we gather the influence score into five bins according to the position of the tokens. We report the average influence scores for each bin.

## A.3 Fine-tuning with Simba

We provide the details for our fine-tuning experiments in Section 4.4.

**Loss design.** The language modeling loss cannot be directly applied to sparsified models because the output logit shape does not match the label shape due to pruning. We apply the language modeling loss only to the available output logits, which are trained to predict the next token in the input sequence. We also add the standard language modeling loss computed using dense forwarding for stable training.

**Training.** We train the model for 400 steps on the MiniPile dataset (Kaddour, 2023), a subset of the Pile dataset (Gao et al., 2020) with similar data distribution. We use AdamW optimizer with a learning rate of 5e-5. We schedule the learning rate with a linear warmup for 10% of the total training steps and cosine learning rate decay for the remaining steps. We randomly select the pruning ratio between 0% and 90% for each sample.

# B  Additional results

The tables below present the full values reported in our experiments.

## B.1  Detailed results for downstream evaluations

Table 4: The full results table corresponding to Figure 3. Bold denotes the best results.

| Model | Scale | FLOPs (x1e12) | Lbd. perp. (↓) | Lbd. acc. (↑) | HS acc. (↑) | PIQA acc. (↑) | Arc-E acc. (↑) | Arc-C acc. (↑) | WG acc. (↑) | Avg. acc. (↑) |
|---|---|---|---|---|---|---|---|---|---|---|
| Simba (ours) | 130m | 0.23 | 31.63 | 36.28 | 35.29 | 64.64 | 49.12 | 24.66 | 49.96 | 43.32 |
| Pythia | 160m | 0.66 | $> 10^2$ | 22.21 | 30.42 | 62.46 | 45.29 | 22.95 | 50.36 | 38.95 |
| Mamba | 130m | 0.53 | 35.81 | 31.84 | 34.88 | 64.36 | 50.76 | 23.72 | 49.80 | 42.56 |
| Simba (ours) | 370m | 0.75 | **16.79** | **45.10** | **46.72** | **69.59** | **58.84** | **28.07** | **55.41** | **50.62** |
| Pythia | 410m | 1.86 | 25.65 | 38.22 | 40.95 | 68.06 | 56.73 | 26.45 | 53.51 | 47.32 |
| Mamba | 370m | 1.51 | 12.34 | 46.61 | 46.46 | 69.59 | 59.05 | 27.47 | 55.80 | 50.83 |
| Simba (ours) | 790m | 1.67 | **10.10** | **51.84** | **54.82** | **72.25** | **65.40** | **32.42** | **58.17** | **55.82** |
| Pythia | 1b | 4.27 | 16.01 | 42.27 | 47.92 | 71.44 | 60.65 | 29.61 | 52.88 | 50.80 |
| Mamba | 790m | 3.24 | 9.44 | 51.33 | 55.12 | 72.47 | **65.49** | 32.76 | 57.54 | 55.78 |
| Simba (ours) | 1.4b | 2.95 | **8.41** | **55.61** | **58.61** | **73.34** | 65.32 | **34.47** | 58.72 | **57.68** |
| Pythia | 1.4b | 6.19 | 10.35 | 49.46 | 53.54 | 70.89 | 65.32 | 33.02 | 57.38 | 54.94 |
| Mamba | 1.4b | 5.60 | 7.31 | 56.43 | 59.60 | 74.05 | 66.50 | 36.18 | 60.14 | 58.82 |
| Simba (ours) | 2.8b | 6.07 | **6.29** | **60.06** | **65.70** | **75.46** | **71.21** | **38.99** | **63.61** | **62.51** |
| Pythia | 2.8b | 12.21 | 7.69 | 54.93 | 60.85 | 74.37 | 68.10 | 36.26 | 60.93 | 59.24 |
| Mamba | 2.8b | 11.31 | 5.80 | 60.85 | 66.73 | 76.12 | 71.97 | 39.93 | 63.69 | 63.21 |

## B.2  Detailed results for perplexity evaluations

Table 5: The full results table corresponding to Figure 4. Bold denotes the best results.

| Model | Scale | FLOPs (x1e12) | Within Context | | | | | Extrapolation | | |
|---|---|---|---|---|---|---|---|---|---|---|
| | | | 0.5k | 1k | 1.5k | 2k | 2.5k | 3k | 3.5k | 4k |
| Simba (ours) | 130m | 0.21 | 3.060 | 3.006 | 2.986 | 2.974 | 2.961 | 2.956 | 2.948 | 2.942 |
| Pythia | 160m | 0.60 | 3.166 | 3.134 | 3.128 | 3.120 | 3.195 | 6.333 | 7.883 | 7.986 |
| Mamba | 130m | 0.48 | 2.943 | 2.917 | 2.907 | 2.899 | 2.899 | 2.900 | 2.901 | 2.905 |
| Simba (ours) | 370m | 0.68 | **2.727** | **2.686** | **2.662** | **2.650** | **2.643** | **2.633** | **2.630** | **2.625** |
| Pythia | 410m | 1.69 | 2.750 | 2.713 | 2.702 | 2.691 | 2.923 | 7.611 | 8.918 | 9.314 |
| Mamba | 370m | 1.38 | 2.645 | 2.616 | 2.607 | 2.600 | 2.599 | 2.601 | 2.604 | 2.617 |
| Simba (ours) | 790m | 1.52 | **2.543** | **2.509** | **2.488** | **2.477** | **2.474** | **2.470** | **2.468** | **2.471** |
| Pythia | 1b | 3.88 | 2.607 | 2.573 | 2.560 | 2.559 | 5.665 | 6.582 | 6.817 | 7.003 |
| Mamba | 790m | 2.94 | **2.486** | **2.457** | 2.445 | 2.437 | 2.436 | 2.443 | 2.458 | 2.497 |
| Simba (ours) | 1.4b | 2.68 | 2.495 | 2.466 | **2.443** | **2.422** | **2.413** | **2.404** | **2.396** | **2.394** |
| Pythia | 1.4b | 5.63 | 2.513 | 2.474 | 2.465 | 2.456 | 3.003 | 6.696 | 7.321 | 7.775 |
| Mamba | 1.4b | 5.09 | 2.389 | 2.363 | 2.354 | 2.348 | 2.347 | 2.352 | 2.358 | 2.372 |
| Simba (ours) | 2.8b | 5.52 | **2.380** | **2.330** | **2.317** | **2.304** | **2.294** | **2.289** | **2.279** | **2.284** |
| Pythia | 2.8b | 11.11 | 2.382 | 2.345 | 2.332 | 2.326 | 6.689 | 7.275 | 7.266 | 7.244 |
| Mamba | 2.8b | 10.28 | 2.281 | 2.254 | 2.242 | 2.235 | 2.236 | 2.254 | 2.314 | 2.547 |

## C   Additional information

### C.1   Compute resources

All experiments are done on RTX-3090 or RTX-2080 GPUs. All FLOPs reported in the paper indicate the computation required for running a single forward pass, so the number of data samples must be multiplied to calculate the amount of computation required for the full experiments. We further acknowledge that the full research project required more computing than the experiments reported in the paper, including the preliminary experiments and failed experiments.

### C.2   License for existing assets

Here, we provide the license for all datasets and models used in our experiments. Apache 2.0 license is applied for the pretrained Mamba models, Pythia models, PG-19 dataset, and WinoGrande dataset. CC BY 4.0 license is applied for the Lambada dataset. CC BY-SA 4.0 license is applied for ARC-Challenge and ARC-Easy datasets. MIT license is applied for the PIQA dataset, HellaSwag dataset, and MiniPile dataset.

## D   Further Discussion and Analysis

### D.1   Inference speed analysis

Throughout the paper, we have used FLOPs as the primary metric for evaluating efficiency. In this section, we present actual inference time measurements to demonstrate that Simba achieves tangible real-world speedups. Specifically, we compare the inference time of Simba and Mamba models at two scales: 790M and 2.8B parameters. To ensure a fair comparison, we base our evaluation on the PyTorch implementation of Mamba's parallel scan operation (Torres-Leguet, 2024), with optimizations to eliminate redundant operations involved in score computation. These modifications allow us to measure the relative gains provided by Simba, and we expect comparable improvements with hardware-optimized implementations, given appropriate optimizations. The results are summarized in Table 6, which reports the average inference time for both Mamba and Simba across various context lengths.

Table 6: **Inference time for various input lengths.** All measurements are taken on 8 RTX 2080 Ti GPUs. We also report the percentage of the speedup.

| Model | Scale | Context Length | | | |
|-------|-------|------|------|------|------|
| | | 1k | 2k | 3k | 4k |
| Mamba | 790m | 0.683 | 1.356 | 2.438 | 2.709 |
| Simba | 790m | 0.598 (14.2% speedup) | 1.115 (21.6% speedup) | 1.724 (41.4% speedup) | 2.164 (25.2% speedup) |
| Mamba | 2.8b | 1.608 | 3.294 | 5.819 | 6.469 |
| Simba | 2.8b | 1.070 (50.3% speedup) | 2.041 (61.4% speedup) | 3.217 (80.9% speedup) | 4.023 (60.8% speedup) |

The results indicate that Simba delivers substantial real-world efficiency improvements, achieving up to an 80% speedup for Mamba-2.8B models. We believe that these gains could be further enhanced by leveraging existing techniques, such as performing token pruning at specific layers rather than at every layer (Liang et al., 2022). This represents a promising direction for future exploration.

### D.2   Application to chunked forwarding

While originally designed for standard prefill scenarios where all tokens are forwarded at once, Simba can be seamlessly adapted to chunked prefill scenarios. In this context, long input sequences are divided into fixed-size chunks for sequential processing. To maintain computational efficiency, token pruning can be performed locally within each chunk.

This can be done by calculating token eviction scores based on the influence of tokens on the last token of each chunk, instead of the last token of the entire input sequence. This localized scoring approach ensures that pruning decisions remain contextually relevant within each chunk, while still preserving the overall efficiency benefits of sparsification. By adopting this method, Simba can achieve similar efficiency gains in chunked prefill scenarios as in standard prefill settings.

### D.3 Analysis for performance improvement under same scale

Table 7: **Performance on downstream tasks.** We evaluate Simba and Mamba models on 6 NLP benchmarks (Lambada, HellaSwag, PIQA, Arc-Easy, Arc-Challenge, and WinoGrande) and report the averaged accuracy. We also report the performance difference between the two approaches with the same model scale. State size corresponds the size of the state tensor corresponding to a single token, at a single SSM layer. All experiment setup follows Table 1.

| Model Scale | 130m | 370m | 790m | 1.4b | 2.8b |
|-------------|-------|-------|-------|-------|-------|
| State Size | 24k | 32k | 48k | 64k | 80k |
| Mamba | 42.56 | 50.83 | 55.78 | 58.82 | 63.21 |
| Simba | 42.75 | 51.01 | 55.80 | 58.21 | 62.86 |
| Difference | +0.19 | +0.18 | +0.02 | -0.60 | -0.35 |

While token pruning is often associated with a decrease in performance compared to dense models, we observed that moderate pruning can enhance performance under certain conditions in Section 4.3. In this section, we conduct a further analysis on this phenomenon.

In Table 7, we compare the averaged accuracy of Mamba and Simba models across 6 NLP benchmarks considered throughout the paper. Noticably, the performance gains are more evident for smaller models with smaller state sizes. We hypothesize that this occurs because smaller state sizes have a limited capacity to retain information and are more prone to forgetting in dense recurrence operations. By introducing highway-like connections through our sparsification method in the upper layers, Simba could mitigate these issues by improving the information flow across distant tokens.

While these results are promising, we believe that further exploration in this direction will be essential to fully understand the underlying mechanisms and to optimize the use of token pruning across a broader range of settings.

### D.4 Comparison with Transformers

Transformers excel at modeling long-range dependencies, as their attention mechanism gives each token direct access to all previous tokens. However, this capability comes at a high computational cost, with complexity growing quadratically with input length.

In contrast, SSMs process sequences with linear complexity by compressing all prior information into a single state and updating it sequentially. While efficient, this design restricts information flow, as tokens can only access the immediate previous state. Over time, dense recurrence operations degrade earlier information, leading to a loss of long-range dependencies.

Simba mitigates the information flow issue in SSMs by sparsifying tokens in upper layers, reducing the number of recurrence steps. This approach alleviates information degradation while preserving the computational efficiency of SSMs. By balancing efficiency and improved long-range information flow, Simba narrows the gap between SSMs and Transformers.

### D.5 Potential failure cases

One potential failure case for Simba arises in scenarios where tokens containing essential information are pruned, leading to a loss of critical context. While this issue would be less severe in comparison to Transformer-based models due to the inherently compressed nature of SSMs, where information from pruned tokens is retained in the states of subsequent tokens, it may still impact the models ability to effectively utilize that information. This limitation could result in performance degradation compared to the dense model, particularly in tasks requiring precise retention of long-range dependencies or complex reasoning.

