# OpenReview forum: "Sparsified State-Space Models are Efficient Highway Networks"
_TMLR — Accepted by TMLR_

### Review · Reviewer_9kSj · 2024-12-16

**Summary Of Contributions:**

This paper introduces Simba, a hierarchical sparsification method for SSM. The main contributions include a token pruning approach that increasingly sparsifies upper layers while preserving lower layers, creating a trapezoidal network structure. The authors develop a new token importance scoring criterion specifically designed for SSMs that measures global token influence by accumulating local recurrences. The work demonstrates that the resulting sparse upper layers act as "highways" that improve information flow across long sequences. Through extensive experiments, the authors show empirical improvements over baseline Mamba models while using equivalent computational resources across multiple NLP tasks and language modeling benchmarks.

**Audience:**

Yes

**Broader Impact Concerns:**

N/A.

**Claims And Evidence:**

Yes

**Requested Changes:**

The authors should demonstrate whether their FLOPs improvements translate to actual training or inference time speedups. This is particularly important for pre-training scenarios where theoretical efficiency gains don't always materialize in practice.

**Strengths And Weaknesses:**

### Strengths
The key strength of this work lies in its direct and intuitive approach. While Transformers rely on KV Cache mechanisms that make skip connections challenging to implement, SSMs can naturally leverage their hidden states to create such pathways. This makes the proposed sparsification approach particularly well-suited for the SSM architecture. The idea itself is straightforward yet effective, demonstrating clear improvements in computational efficiency as measured by FLOPs.
### Weaknesses
The primary weakness centers on the reliance on FLOPs as the main efficiency metric. As someone with experience in pre-training large models, I find this concerning because FLOPs improvements often fail to translate into actual training speed gains. Many factors beyond raw FLOPs affect real-world training performance, including memory access patterns, hardware utilization, and implementation efficiency. The paper would be strengthened by demonstrating whether these theoretical FLOPs advantages can actually deliver practical speedups in training or inference scenarios.

Furthermore, while the authors propose a token pruning criterion based on measuring global influence through local recurrences, they don't adequately address potential failure cases. For instance, in scenarios where critical information is distributed across multiple tokens rather than concentrated in individual "important" tokens, their pruning approach might discard essential contextual information. This could be particularly problematic for tasks requiring complex reasoning or long-range dependencies.

---

> ### Author Response · Authors · 2025-01-22
> **Response to Reviewer 9kSj**
>
> Dear reviewer 9kSj,
>
> We sincerely appreciate your efforts in reviewing our manuscript. We respond to each comment in the following content. We carefully incorporated the discussions into the revised manuscript. We highlighted the revised contents in blue for your convenience to check.
>
> ## Demonstration of real-world speedup.
>
> We acknowledge your concern that FLOPs alone may not always reflect practical efficiency gains. To address this, we conducted additional experiments to measure real-world inference time, which are detailed in Appendix D.1. These experiments show that Simba achieves up to 80% improvement in inference speed compared to Mamba for 2.8B models. This demonstrates that the theoretical improvements indeed translate to tangible performance benefits in practice. We hope this provides clarity and strengthens the validity of our claims regarding computational efficiency.
>
> ## Potential failure cases.
>
> Thank you for pointing out the importance of discussing potential failure scenarios. We have included a more detailed analysis in Appendix D.5, focusing on the potential limitations of Simba in scenarios where pruning may discard critical information. While we expect SSMs to be more robust to this issue compared to token pruning for Transformer models due to their compressive nature—where information from pruned tokens is retained in subsequent token states—such cases can still impact performance.

---

### Review · Reviewer_hcUQ · 2025-01-08

**Summary Of Contributions:**

In this work, the authors propose Simba, which is  a simple yet effective method to sparsify SSMs through token pruning. Here, the main motivation is that since SSMs generate the next states based on the immediate previous states and current input, compressing all information into one state will result into redundancy. Therefore, the authors will prune more tokens for upper layers, which  behave like highways. To determine which token to be pruned, the influence function is used, which measures the importance of token $x_t$ by subtracting the final token output with and without this token. Experimental results show the effectiveness of proposed method.

**Audience:**

Yes

**Claims And Evidence:**

Yes

**Requested Changes:**

1. Add results about improvement in practical efficiency to show how much Simba can accelerate prefilling
2. Discuss about the chunkwise prefilling, and clarify how to prune tokens within and among chunks.
3. Add analyses for the better performance of Simba compared to Mamba even using less FLOPs.
4. Update Table 2 with comparison among models with the same scale.

**Strengths And Weaknesses:**

Strengths:

1. The paper is well-written.
2. The motivation of Simba is very clear.
3. It shows better performance when using same flops in comparison to Mamba. In addition, by dropping redundant information during inference, it even increase performance despite using

Weaknesses:

1. Efficiency: It seems this method can only reduce prefilling time, while during generation we cannot drop tokens as we need to compute the current state to predict the next token. Moreover, when it comes to prefilling, we will compute it in a chunkwise manner. Based on the current implementation, we will only prune tokens in each chunk, which should be computed in parallel, therefore it is not clear about the practical improvement in training speed by using less FLOPs. It is difficult for us to completely prune a whole block based on the current implementation explained in the paper. Based on these, I am concerning about the practical speedup and suitable settings.
2. Performance: It is very impressive that Simba can improve performance from Mamba when using less FLOPs, however, the authors do not provide a clear explanation of this part. Additionally, I would expect the authors to mainly focus this improvement in the motivation.
3. Experiments: In Table 2, why Mamba and Simba with different scales are compared. This is a little strange.

---

> ### Author Response · Authors · 2025-01-22
> **Response to Reviewer hcUQ**
>
> Dear reviewer hcUQ,
>
> We sincerely appreciate your efforts in reviewing our manuscript. We respond to each comment in the following content. We carefully incorporated the discussions into the revised manuscript. We highlighted the revised contents in blue for your convenience to check.
>
> ## Efficiency gains in chunked prefilling scenarios.
>
> We appreciate your comment regarding the efficiency of Simba in chunked prefilling scenarios. While Simba was originally designed for standard prefilling, it can indeed be adapted for chunked prefilling, where long inputs are segmented into fixed-size chunks for sequential processing.
>
> In such scenarios, token pruning can be performed within each chunk. Specifically, the token eviction scores are calculated based on their influence on the last token of the respective chunk, rather than the last token of the entire input. This localized pruning ensures computational efficiency while maintaining the model's performance within each segment. We have added a detailed explanation of this approach in Appendix D.2 to clarify how Simba can be adapted to chunked prefilling scenarios.
>
> ## Improvements in practical efficiency.
>
> To further address your concern about practical efficiency, we conducted additional experiments to evaluate real-world speedups achieved by Simba. These experiments focused on measuring inference time during prefilling for models of various scales. The results, detailed in Appendix D.1, demonstrate that Simba achieves up to 80% improvement in inference speed compared to Mamba for 2.8B models. This underscores Simba’s practical efficiency gains and highlights its potential for real-world applications.
>
>
> ## Further analysis for performance improvements at the same scale.
>
> While token pruning is often associated with a decrease in performance compared to dense models, we observed that moderate pruning can enhance performance under certain conditions. To investigate this phenomenon further, we conducted an analysis presented in Appendix D.3.
>
> Our findings suggest that the observed improvement is related to the model's scale and state size. Specifically, smaller models with smaller state sizes appear to benefit more from pruning. We hypothesize that this occurs because smaller state sizes have a limited capacity to retain information and are more prone to forgetting in dense recurrence operations. Thanks to the highway connections created through our sparsification method in the upper layers, Simba could mitigate these issues by improving the information flow. While these results are promising, we believe that further exploration in this direction will be essential to fully understand the underlying mechanisms and to optimize the use of token pruning across a broader range of settings.
>
> ## Comparing Mamba and Simba with different scales.
>
> Thank you for highlighting the confusion in Table 2. To clarify, the original table was designed to showcase performance differences between training-free and fine-tuned models, with Mamba included as a reference.
>
> In the previous version, we included the 130M model performance to emphasize how Simba’s sparsification approach outperforms smaller dense models with comparable computational budgets. However, we understand that this presentation may have caused confusion.
>
> To improve clarity, we have revised Table 2 to include a direct comparison between models of the same scale, specifically adding the performance and FLOPs of the Mamba-370M model. This update aims to provide a clearer and more accurate representation of Simba’s advantages, highlighting its ability to balance efficiency and performance effectively.

---

### Review · Reviewer_QBdD · 2025-01-09

**Summary Of Contributions:**

The authors propose a method to sparsify state space models (SSM), in particular a recent improvement 'Mamba'. This has two benefits. One the one side it reduces the computational complexity of SSMs further. On the other side, it allows better performance on sequences with long contexts, as they are more efficiently propagated through the network. The sparsification that is proposed is hierarchical, it prunes more weights 'upper' layers of the network, as they are shown to be more similar to each other than the lower layers. The filters that are pruned are selected based on how much the output of a model changes when they are removed. Experiments on various NLP benchmarks are performed, and under the same computational budget, the proposed Simba outperforms Mamba. Additional experiments show that the fewer tokens in the upper layers indeed improve the information flow between distant input tokens. A final ablation study verifies that the same results are not reached when using random or uniform pruning.

**Audience:**

Yes

**Claims And Evidence:**

Yes

**Requested Changes:**

**Necessary**

* A clearer indication that the computational cost that is concerned only relates to inference, and that training costs are higher, should be included.
* A clearer discussion of the data that is used to prune the network is necessary (but not all the ablations I proposed, although they would improve the paper).

** Optional **

Everything else I mentioned in the above weakness section.

**Strengths And Weaknesses:**

**Strengths**

The paper clearly introduces a hypothesis, verifies and improves it. The hypothesis is that fewer tokens are necessary in the upper layers because the information there is less influenced by the last new token, which is clearly show in Section 3.1. A reasonable solution is proposed, and it is verified that this solution actually does what is intended (results in Figure 4 and 5), apart from only on NLP benchmarks. This thorough approach is highly appreciated.

**Weaknesses**

There are a few smaller things that I think can improve the paper further.

* The paper is concerned about a fixed computational budget, but if I understand everything correctly, this is only considering the computational cost of inference. The training cost of a Simba model with the same inference cost as a Mamba model still required a larger training cost. I think this could be made more explicit when talking about the computation cost throughout the paper.
* The scores in Table 1 are not that different. As far as I could tell, the input data of the pruning can be the only source of variation in the pruning. It is unclear which data and how much is used for the pruning. Both may be a source of variability in the pruning and eventual results. It implies some important questions: is the pruning asymptotic and are the same filters pruned if enough data is used, or is there always some variability? Does the data source that is used matter? Is it important that the data that is used to prune is close to the eventual test data? If possible, it wouldn't be bad to provide standard deviations in Table 1. I understand that it is not the goal of this method to have higher accuracy than Mamba, but right now this is claimed in the paper and given that there likely is some variability, this is not sufficiently backed up.



* In the introduction there is the reasoning step that more redundancy is necessarily a result of more global information. I don't think this is the only reason that redundancy can happen, and it is not a necessary argument for the method. I think simply observing that there is more redundancy is enough to motivate the method, with speculating on the 'kind' of information.
* SSMs and especially Mamba are proposed as an less computational complex alternative for transformer models. It is not absolutely necessary, but a comparison between the downsides and benefits of SSMs and transformers and how Samba possibly resolves these would be interesting.

---

> ### Author Response · Authors · 2025-01-22
> **Response to Reviewer QbdD**
>
> Dear reviewer QBdD,
>
> We sincerely appreciate your efforts in reviewing our manuscript. We respond to each comment in the following content. We carefully incorporated the discussions into the revised manuscript. We highlighted the revised contents in blue for your convenience to check.
>
> ## Clearer indication of training costs.
>
> We have clarified in the revised manuscript that the computational efficiency discussed in our work primarily focuses on inference. While Simba is designed to optimize inference efficiency, we acknowledge that it incurs higher training costs due to the larger model size required during training. To address this, we revised the Introduction, Related Work, and Method sections to explicitly state that our method targets inference efficiency. Additionally, we added a discussion in the Limitations section to acknowledge the higher training costs.
>
>
> ## Clarification for the data used for pruning.
> Simba dynamically determines pruning decisions during inference based solely on the input provided at that time. No external pruning data is used, and therefore the pruning process is deterministic. As a result, pruning patterns do not vary, and the results in Table 1 are not subject to variability introduced by pruning data. We have revised the Method section to clarify that pruning decisions depend entirely on the given input and are deterministic.
>
> ## Unnecessary claims regarding the source of redundancy
>
> We appreciate your feedback regarding the reasoning flow of the Introduction section.
> We have revised the Introduction to remove the claim that redundancy in upper layers necessarily arises from more global information. Instead, we focus on the observed redundancy in these layers, which is sufficient to motivate our sparsification approach.
>
> ## Comparison between Transformer models
> We appreciate your suggestion to include a comparison between Transformers and SSMs. Transformers are highly effective at capturing long-range dependencies due to their attention mechanism but suffer from quadratic computational complexity with respect to input length. SSMs, in contrast, offer linear computational complexity but face challenges such as restricted information flow and degradation of long-range dependencies caused by dense recurrence operations.
> Simba addresses these limitations by sparsifying the upper layers of SSMs, improving long-range information propagation while maintaining linear complexity. To provide additional context, we included a detailed discussion in Appendix D.4, which compares the trade-offs between Transformers and SSMs and highlights how Simba mitigates SSMs’ limitations.

---

### Decision · Action_Editor_2PnG · 2025-02-10

**Recommendation:** Accept as is

**Comment:**

This paper introduces Simba, a hierarchical token pruning method for state-space models (SSMs) that sparsifies upper layers more aggressively, enhancing both computational efficiency and long-range information flow. While initially focused on FLOPs as an efficiency metric, the authors convincingly addressed reviewer concerns by demonstrating substantial real-world inference speedups and clarifying increased training costs and adaptability to chunked pre-filling. They also added a relevant discussion regarding failure scenarios where information might be more evenly distributed. The paper is recommended for acceptance as is; it offers a novel, well-supported, and empirically validated approach, making a strong contribution to sequence modeling.

**Audience:**

Yes, this paper will interest members of the TMLR community.

**Claims And Evidence:**

Yes, the claims are supported by clear and convincing evidence.